# Does the threat of COVID-19 modulate automatic imitation?

James W. Roberts⬤*, Simon J. Bennett

Brain & Behaviour Research Group, Research Institute for Sport & Exercise Sciences (RISES), Liverpool John Moores University, Liverpool, United Kingdom

* J.W.Roberts@ljmu.ac.uk

## Abstract

The tendency to involuntarily imitate the actions of others (automatic imitation) can be modulated by social affiliative cues. Here, we explored whether the disruption to our social lives caused by the COVID-19 pandemic may subsequently influence automatic imitation. Three groups were initially presented a sentence comprehension task that featured either neutral (control), safe or unsafe primes to COVID-19 infection. They then completed an automatic imitation task, where a numeric cue was presented alongside apparent motion of an index or middle finger, which was either compatible or incompatible with the required response. Reaction times were longer for the incompatible compared to compatible trials, and thus demonstrated automatic imitation. However, there was no influence of the primes indicating that automatic imitation was unaffected by the risk of COVID-19. The potential theoretical explanations and practical implications of pathogen avoidance and social bonding incentives are discussed with reference to pandemic events.

**Data Availability Statement:** All data files are available from Open Science Framework (https://osf.io/a8xzr/).

**Funding:** The authors received no specific funding for this work.

## Introduction

It is widely known that humans tend to mimic/imitate each other [1–3]. This feature is most succinctly and perhaps best captured by the automatic imitation paradigm; that is, the tendency to delay rapid manual responses (e.g., index finger flexion) to observed stimuli that coincidentally feature images of an incompatible (e.g., middle finger flexion) as opposed to compatible (e.g., index finger flexion) manual gesture [4–6] (for a review, see [7, 8]). This behaviour is suggested to unfold following the mapping of observed actions onto the observers' own action system, which can manifest as facilitation or interference for compatible and incompatible stimuli, respectively [9, 10]. A potential neural substrate underpinning these findings has been offered following the discovery of mirror neurons within the premotor cortex of the macaque (F5), which respond equally to the observation and execution of actions ("monkey see, monkey do") [11]. Indeed, a similar mirror mechanism has been identified within humans using measures of corticospinal excitability [12–14] and neural hemodynamic responses (also known as the Action-Observation Network (AON)) [15–18].

For the past decade, research within this area has broadly focused on the top-down modulation of automatic imitation, where surrounding contextual priors may either increase or

**Competing interests:** The authors have declared that no competing interests exist.

decrease the magnitude of imitation. A key finding is that individuals either increase or decrease their imitation depending on whether the situation lends itself to imitation (e.g., pro-social), or is deemed inappropriate (e.g., anti-social), respectively [19–21]. For example, automatic imitation is reported to increase following the prior exposure to pro-social cues (e.g., friend), while it decreases following the presence of anti-social cues (e.g., alone) [22, 23] (for similar findings, see [24–27]) (cf. [28, 29]). This modulation is suggested to unfold courtesy of a separate, but interacting, neural network that is synonymous with social cognition, including the anterior medial frontal cortex (aFMC) and temporoparietal junction (TPJ) [30–33].

Given the strong influence of contextual priors, and more specifically social norms, on automatic imitation, it is relevant to consider the impact of the COVID-19 pandemic. Indeed, the onset of the pandemic and the measures imposed to mitigate the spread of the virus (e.g., self-isolation, social distancing, mask wearing) have led individuals to challenge their pre-existing relationship with the world around us and how we interact with one another (e.g., [34–38]). Therefore, any pre-existing urge to bond with others could be in direct conflict with our need to survive [39, 40]. In research on pathogen avoidance, this is reflected in the notion of the behavioural immune system (BIS) [41], which suggests that as well as the physiological immune response required to directly combat disease, humans have evolved in such a way to avoid contracting diseases in the first place. To elucidate, individuals with a dispositional tendency or situational prime to infectious disease can generate avoidance movements to maximize distance between agents, higher ratings on a scale of perceived vulnerability to disease, and an overperceived bias toward disease-related cues (e.g., obesity, age) [42, 43].

While there has been some scepticism surrounding whether BIS can offer a viable explanation or understanding of avoidance behaviour during the COVID-19 pandemic [44], there is early evidence to suggest that individuals who express greater concern toward the pandemic and implement more proactive avoidance behaviours, tend to rate themselves more highly on scales of germ avoidance and disgust sensitivity; two key components of BIS [45, 46]. Therefore, it is possible that a prime related to COVID-19, and the subsequent attempts to avoid this disease may influence our relationship with other people, and more specifically, the social communicative or affiliative tendency to imitate others [1, 19, 20].

To this end, the present study explored the influence of the COVID-19 pandemic on the magnitude of automatic imitation. Participants were presented with a situational prime condition that was either neutral, safe or unsafe with respect to the measures imposed for combating COVID-19 in the United Kingdom (UK). The neutral primes represented a control condition that did not include any content related to the pandemic, whereas the safe and unsafe primes contained information on measures that were either averse (e.g., regular mask wearing, video conferencing) or tolerant (e.g., no mask wearing, face-to-face interaction) to the threat of COVID-19, respectively. To elucidate, the safe primes referred to measures designed to mitigate the risk of infection, which should have no impact upon typical social affiliative behaviour, thereby manifesting in typical or baseline automatic imitation. Conversely, the unsafe primes referred to a heightened risk of infection, which should increase the need to create distance between individuals and diminish the incentive for social affiliation, thereby manifesting in reduced automatic imitation [39]. To further explore the nature of this automatic imitation; specifically, whether it was a product of attentional processes within early perception and/or inhibitory processes within late response production, we further separated reaction time into fast and slow time bins. That is, if the automatic imitation is attributed to early perception, then we would anticipate these effects to be greater within the fast time bins, although if they are attributed to late response production, then we would anticipate it happening within slow time bins (for a conceptual framework, see [47, 48]). Because automatic imitation generally unfolds within more prolonged reaction times [49], while the top-down modulation of

automatic imitation is primarily attributed to late inhibitory processes [50], we predicted that any differences in automatic imitation to feature within the slow as opposed to fast time bins.

## Method

### Participants

There were 64 undergraduate students that initially provided consent online, although only 59 of them went on to successfully complete the study (age range = 18–30 years, right-handed n = 50, left-handed n = 9, men n = 40, women n = 19). Participants were randomly allocated to one of 3 groups that were each characterised by their unique prime content (control n = 18, safe n = 21, unsafe n = 20) (for further details, see *Tasks and Procedure*). Data collection followed three separate calls for participants (February 2021, November 2021, February 2022), which coincided with a nationwide lockdown and mass vaccination programme in the UK, and ceased entirely when nearing the end of nationwide restrictions. The subsequent sample size aligned with previous other studies on automatic imitation (e.g., [22, 23]). The study was designed and conducted in accordance with the Declaration of Helsinki, and approved by the Liverpool John Moores University Research Ethics Committee.

### Tasks and procedure

The study was designed and conducted using Gorilla Experiment Builder (https://gorilla.sc/). As an online behavioural experiment platform, Gorilla enables participation within more remote and convenient settings (e.g., home office) that can help in the recruitment of a larger and more representative sample, while still retaining a high temporal precision for any user input required in stimulus-response designs [51, 52].

The situational primes were issued in the form of a sentence comprehension task, which is aligned with a number of previous studies that have explored prime-induced top-down modulation of automatic imitation (e.g., [22, 23, 26, 49, 53]). Participants received incomplete sentences, which they were instructed to complete in order to form grammatically correct sentences as part of a language comprehension test (S1 Text). Each of the sentences had 2 missing words with 4 available words to freely choose from. Participants were informed that the sentences were adapted from newspaper headlines taken over the course of the past 18 months. These headlines were in fact created by the first author, although they did loosely reflect real-world events. Because of differences in the time of year when participants were tested (see *Participants* sub-section), it is possible that there was some subtle variation in the events reported within the so-called headlines that comprised each of the primes, and those reported within the real-world media. However, we do not expect participants to have had a sufficiently precise recollection of the real-world events such that they would become consciously aware of this potential discrepancy.

The nature of the sentences differed depending on the intended prime content for each group. That is, the control group received solely neutral primes (e.g., "Marine <u>biology</u> becoming one of the <u>fastest</u> growing areas of study"), while the safe and unsafe groups received primes that were either averse (e.g., "Video <u>conferencing</u> officially substitutes the <u>boardroom</u>") or tolerant (e.g., "<u>Thousands</u> go <u>back</u> to the shops") to the threat of COVID-19, respectively. In an attempt to control for other potentially confounding influences, the primes were piloted on a separate sub-set of individuals (n = 6), which revealed differences in valence/pleasantness and arousal, but crucially, no difference in sociability (S2 Text). In order to help conceal the true nature of the study and avoid any subjective bias, half of the sentences for the safe and unsafe groups were from the neutral primes (for similar procedures, see [22]). In sum, there

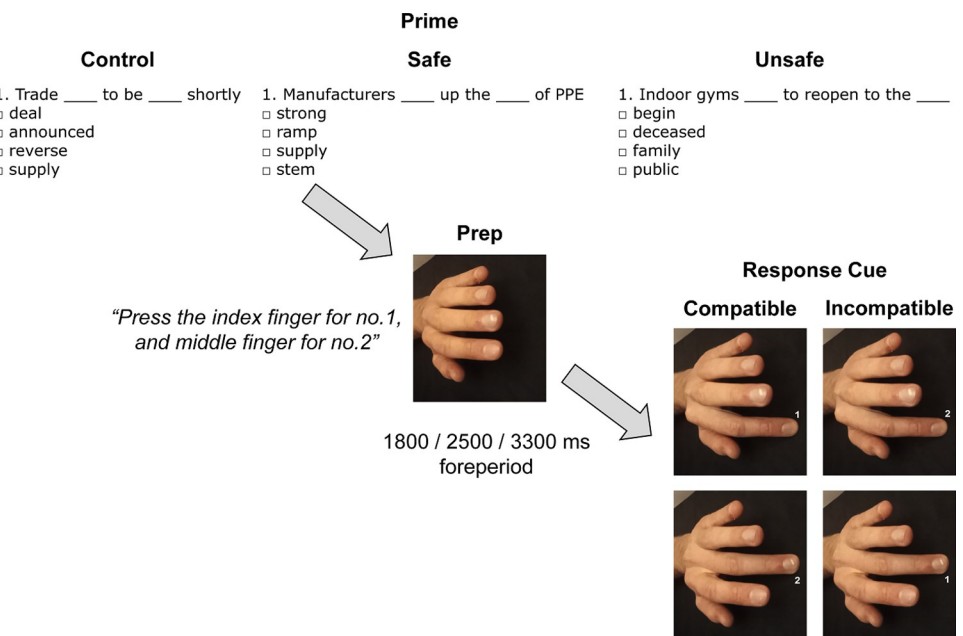

**Fig 1. Outline of the tasks and procedure.** Participants were first assigned one of the prime conditions (control, safe, unsafe) before attempting the stimulus-response task, where the background hand stimuli were either compatible or incompatible with the cued response.

were a total of 18 sentences for each group to complete: 18 neutral for the control group, 9 safe and 9 neutral for the safe group, 9 unsafe and 9 neutral for the unsafe group.

Upon completion of the sentence comprehension task, participants immediately began the automatic imitation task. Each trial began with the presentation of a neutral hand stimulus including a photograph of an outstretched human left hand with the fingers slightly raised above the black surface background (Fig 1). While all images were originally captured in a prone position, the hand stimuli were reoriented 90 degrees anti-clockwise in order isolate any compatibility effects to the anatomical configuration of the hand gesture, and not the mere spatial characteristics (for similar procedures, see [49, 53–55]). Participants would then indicate that they were ready to continue the trial by simultaneously using their index and middle fingers of their right hand to press-and-release the "V" and "B" keys of the keyboard, followed by their hand remaining in the neutral position above the keys until the response cue appeared. Following a random foreperiod (1800, 2500, 3300 ms), the neutral hand stimulus was replaced with another hand stimulus that acted as the response cue. This hand stimulus showed either the index or middle finger being flexed such that it was pressed down closer to the surface, thereby invoking a sense of apparent finger motion. Overlaid on the hand stimulus was a small number "1" or "2" presented in between the index and middle fingers. The number "1" signalled participants to use the index finger to press down on the "V" key, and number "2" signalled participants to use the middle finger to press down on the "B" key. Depending on the observed configuration of the fingers and identity of the numeric response cue, the stimuli were either compatible or incompatible with the cued response. That is, compatible trials were those that featured an image of the index or middle finger being flexed, combined with a cue to press down with the index ("1") or middle ("2") finger, respectively. Incompatible trials were those that displayed an image of the index or middle finger being flexed, combined with a cue to press down with the opposing middle ("2") or index ("1") finger, respectively. There were a total of 40 trials comprising of 20 compatible and 20 incompatible, which were

presented completely at random. This number was chosen for the sake of participant convenience, and thus facilitated adherence to the study, while remaining consistent with the procedures adopted within previous other studies (e.g., [23]). Prior to the experiment for real, participants had the opportunity to attempt 8 practice trials including feedback so they could quickly come to terms with the task requirements.

### Dependent measures and data analysis

The dependent measure of interest was reaction time, which was defined as the time difference between the initial presentation of the response cue and subsequent key press response. Prior to analysis, we removed reaction times that were <100 ms (i.e., false starts), and any that were >2500 ms (i.e., delayed response). To avoid any potential trade-off between speed and accuracy, we also removed any trials that were responded to in error (e.g., compatible response within an incompatible trial). Data are available via the Open Science Framework (https://osf.io/a8xzr/).

The participant mean reaction times were initially analysed using a two-way mixed-design ANOVA including the between-measures factor Prime (control, safe, unsafe), and within-measures factor Compatibility (compatible, incompatible). In an attempt to further explore the compatibility effect—synonymous with automatic imitation—we also decomposed the reaction times into time bins [49, 50]. Specifically, we categorised the reaction times from each trial as either fast or slow according to a median split of the data for each individual participant within each condition. To this end, we analysed the participant mean reaction times using a three-way mixed-design ANOVA including the between-measures factor Prime (control, safe, unsafe), and within-measures factors Compatibility (compatible, incompatible) and Time Bin (fast, slow). Because this study was primarily concerned with the magnitude of automatic imitation, any potential factorial interactions were interpreted based on the size of the mean difference between the two levels of compatibility; that is, compatible and incompatible trials. Significance was declared at $p < .05$ (two-tailed). Effect sizes related to statistical outcomes were captured using partial eta-squared ($\eta_p^2$).

Additionally, in order to corroborate the aforementioned frequentist statistical approach and avoid any general issues with null hypothesis testing [56, 57], we conducted Bayesian ANOVAs using JASP (v.0.14.1). Specifically, for the analysis of effects, we compared only matched models to isolate any factorial interactions without main effects [58]. Therein, we reported on the Bayes Factor; that is, an odds ratio pertaining to predictive power of the alternative hypothesis over the null, with the former taken as the numerator and the latter as the denominator ($BF_{10}$). Thus, a value <1 favours the null hypothesis, and a value >1 begins to favour the alternative hypothesis (see [59, 60]).

### Results

Two participants were removed due to a failure to follow the instructions having incorrectly responded to nearly all the incompatible trials (89–95% errors) (control $n = 18$, safe $n = 20$, unsafe $n = 19$). Of the remaining participants, all successfully completed the sentence comprehension task that was related to the situational prime condition (range = 0–8 errors (out of 36 words)). A subsequent posteriori sensitivity analysis conducted in G*Power (v.3.1.9.4) [61, 62], including the desired parameters of $\alpha = .05$ and $1\text{-}\beta = .80$, revealed the minimum estimated effect size that could be positively detected was $\eta_p^2 = .043$ (f = .21). Inversely, a posteriori power analysis using the same platform indicated that an $\alpha = .05$ and $\eta_p^2 = .08$ (f = .29) (see Prime x Compatibility statistical interaction) provided statistical power of $1\text{-}\beta = .97$. Only 3.13% of trials on the automatic imitation task failed to qualify due to the time and/or error in

responses, and were subsequently removed prior to the analysis (control no. trials: compatible-fast $M = 9$, $SD = 1.42$, compatible-slow $M = 9$, $SD = 1.46$, incompatible-fast $M = 9$, $SD = 1.20$, incompatible-slow $M = 9$, $SD = 1.24$; safe no. trials: compatible-fast $M = 10$,, $SD = .57$, compatible-slow $M = 10$, $SD = .52$, incompatible-fast $M = 9$, $SD = .75$, incompatible-slow $M = 9$, $SD = .75$; unsafe no. trials: compatible-fast $M = 10$, $SD = .42$, compatible-slow $M = 10$, $SD = .32$, incompatible-fast $M = 10$, $SD = .58$, incompatible-slow $M = 9$, $SD = .77$).

Reaction time showed no significant main effect of Prime, $F(2,54) = .95$, $p = .40$, $\eta_p^2 = .03$, $BF_{10} = .55$, although there was a significant main effect of Compatibility, $F(1,54) = 22.43$, $p < .001$, $\eta_p^2 = .29$, $BF_{10} = 1010.31$, which indicated that participants were slower to respond to the incompatible ($M = 570$ ms, $SE = 19$) compared to compatible ($M = 526$ ms, $SE = 14$) trials. There was no significant Prime x Compatibility interaction, $F(2,54) = 2.35$, $p = .11$, $\eta_p^2 = .08$, $BF_{10} = .76$.

Further inspection of the reaction times as a function of time bins revealed that for the within-participant means, there was no significant main effect of Prime, $F(2,54) = .94$, $p = .40$, $\eta_p^2 = .03$, $BF_{10} = .46$, although there was a significant main effect of Compatibility, $F(1,54) = 23.14$, $p < .001$, $\eta_p^2 = .30$, $BF_{10} = 4009.16$, and Time Bin, $F(1,54) = 199.57$, $p < .001$, $\eta_p^2 = .79$, $BF_{10} = 4.201 \times 10^{38}$. These effects were superseded by a significant Compatibility x Time Bin interaction, $F(1,54) = 5.27$, $p = .03$, $\eta_p^2 = .09$, $BF_{10} = .42$, whereby the compatible-incompatible difference was more pronounced within the slower compared to faster reaction time bins ($t(56) = 2.35$, $p = .02$) (Fig 2 and Table 1). There were no other significant factorial interactions (Prime x Compatibility: $F(2,54) = 2.33$, $p = .11$, $\eta_p^2 = .08$, $BF_{10} = .65$; Prime x Time Bin: $F(2,54) = .32$, $p = .73$, $\eta_p^2 = .01$, $BF_{10} = .12$; Prime x Compatibility x Time Bin: $F(2,54) = 1.22$, $p = .30$, $\eta_p^2 = .04$, $BF_{10} = .20$).

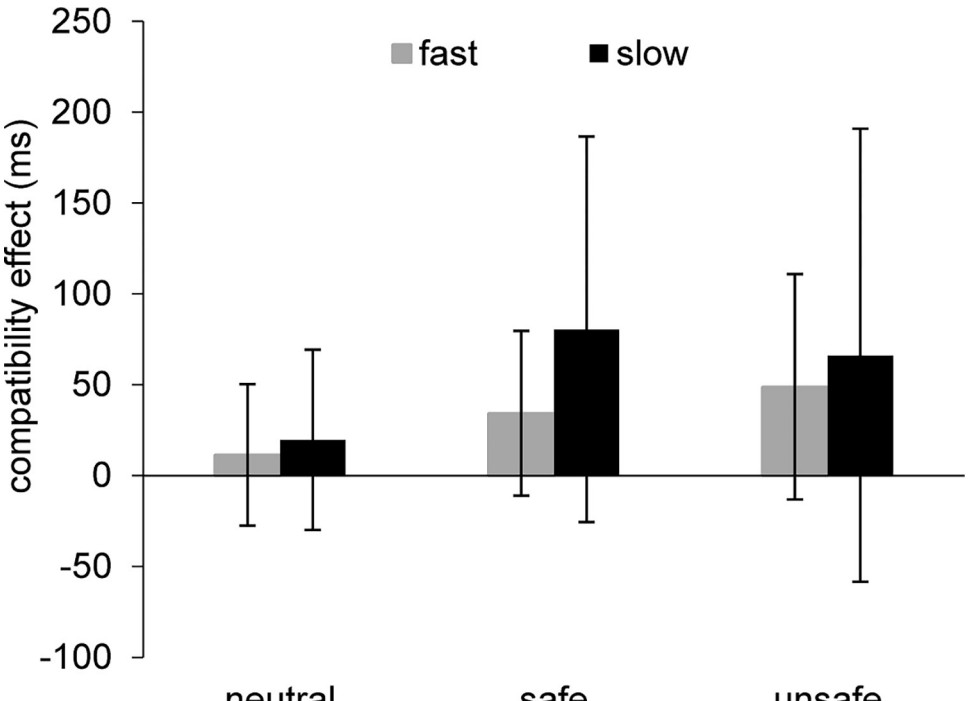

**Fig 2. Mean participant compatibility effect (compatible-incompatible difference with positive values being synonymous with automatic imitation) (±SD) as a function of prime group (neutral, safe, unsafe) and reaction time bin (fast, slow).**

**Table 1. Mean participant reaction time (ms) (±SE) within compatible and incompatible stimulus-response trials as a function of prime group and reaction time bin (fast, slow).**

|  | Fast | | Slow | |
|---|---|---|---|---|
|  | Compatible | Incompatible | Compatible | Incompatible |
| Neutral | 431 (±18) | 443 (±16) | 595 (±33) | 614 (±32) |
| Safe | 452 (±16) | 486 (±24) | 588 (±22) | 668 (±40) |
| Unsafe | 459 (±20) | 508 (±28) | 632 (±41) | 698 (±52) |

## Discussion

As with the natural tendency to mimic others during our daily interactions [1, 27], automatic imitation is highly sensitive to the contextual priors including the social setting [22, 23]. To elucidate, cues to affiliate or distance oneself from others may subsequently increase or decrease the magnitude of automatic imitation, respectively. Meanwhile, dispositional tendencies or situational primes to infectious disease can cause avoidance behaviours (courtesy of the BIS), which runs counter to any potential urge to affiliate with others [39, 40]. In the current study, we considered whether the urge to affiliate, and thereby automatic imitation, is adversely affected by primes to infectious disease; namely, the emergence of COVID-19.

Consistent with previous studies on automatic imitation (for a review, see [7, 8]), we found a significant compatibility effect, whereby the participants' mean and variability in reaction time were larger for the incompatible compared to compatible trials. This effect is suggested to highlight the mapping of observed actions onto the observer's own action system [9, 10] (see also, [63, 64]). What's more, we found that the compatibility effect was most prevalent within the slower as opposed to faster reaction time bins. Such an effect would appear to indicate an influence primarily within the late inhibitory stage of processing rather than early perception [49, 50]. This so-called late stage has been linked to the notion of top-down modulation, where individuals must try to distinguish between the observed actions of others (e.g., index finger flexion) and their own intended action (e.g., middle finger flexion) [30–32, 47, 48]. Therein, they may inhibit any initial influence of automatically mapping observed actions onto their own action system, and instead generate an alternative action that is demanded by the task. While this process clearly ensures that we do not involuntarily mimic/imitate within most of our daily interactions (for examples of neuroatypical cases, see [30, 65–68]), it can come at the expense of time and resources [69, 70].

Importantly, and contrary to our initial hypothesis, the strong automatic imitation effect in the current study did not systematically differ between the different prime groups, nor according to Bayes Factor analyses did the observed data more closely fit the alternative research hypothesis compared to the null hypothesis. Indeed, it was originally suggested that the perceived threat of disease following the unsafe primes, which often elicit avoidance behaviours [42, 43] (for examples related to the COVID-19 pandemic, see [45, 46, 71]), could reduce automatic imitation. One possibility is that despite the increased risk of infection portrayed by the unsafe primes, the inclusion of content on upholding social norms within these primes (e.g., "Visits to the supermarket increase as lockdown hits") may have offset any potential reduction in automatic imitation [20, 22, 23] (see also, [19, 21]). Sensitivity to such social cues appears to unfold not only within ecologically valid face-to-face settings that are more regularly associated with everyday mimicry [1], but also computer-based stimulus-response paradigms as within the present study [20, 22, 23, 49].

With this in mind, we may question how it is that in the absence of these social cues, such as within the safe primes (e.g., "Larger firms to retain measures of remote working")

participants continued to exhibit automatic imitation. Indeed, if the fore mentioned influence of social cues modulates automatic imitation, then it stands to reason that the attempts of the safe primes to withdraw from daily social interactions could have alternatively reduced automatic imitation. However, it could be that despite attempts to physically distance from others, there were still sufficient cues to continue to socially engage; albeit using alternative means (e.g., video-conferencing, virtual party apps, online gaming, etc) (cf. [71, 72]). In this regard, it would be premature at best to consider social distancing as an anti-social cue; particularly within the present context where distancing is borne out of necessity and stereotypical social behaviours have to navigate the challenges of the COVID-19 pandemic. Consistent with this logic is evidence that some individuals during the course of the pandemic actually experienced less negative social affect as a result of being able to virtually communicate and interact with others [37]. This feature may have been compounded by a sense of unity or togetherness because the pandemic can be seen as a common problem faced by everybody, which perhaps strangely brings individuals closer together (e.g., weekly 'Clap for Our Carers', livestreaming events, etc) [73, 74].

Aside from the above theoretical explanations, it is possible that the use of situational primes did not fully exert their influence or had less of an effect on young adult participants of the current study. For instance, recent research shows that while age is negatively correlated with perceived risk or vulnerability to COVID-19 (i.e., contracting disease), it is positively correlated with risk severity (i.e., consequences of disease) [75, 76] (see also, [45, 46]). Likewise, there are a number of other individual differences that have been strongly linked with views on the COVID-19 pandemic (e.g., conspiracy theories; [77]). These findings are consistent with research on how matters such as political ideology [78], prejudicial views and social biases [42, 79, 80] and cognitions/decision-making [81] are all linked to dispositional tendencies to infectious disease in general. Currently, we can only suggest that based on pre-experimental ratings of valence/pleasantness and arousal (S2 Text), a similar group of young adult participants did generally have negative feelings toward the pandemic, and thus perceived the primes in the expected direction. Future work should more closely consider the impact of factors surrounding individual differences or dispositional tendencies when investigating pathogen avoidance behaviours including, but not limited to, imitation.

It is also possible that the data collected at separate timepoints over the course of a 12-month period may have influenced the level of response toward the situational primes. To elucidate, data collection initially coincided with high levels of disease transmission and hospitalisation, national lockdown and limited vaccination with access permitted to only to the most vulnerable of individuals across the UK (February 2021). Additional data were subsequently collected at a time when there were fewer restrictions and unlimited access to vaccines for all ages (November 2021, February 2022). With this in mind, we undertook further inspection of the timepoints for data collection with each of the participants, which revealed no influence of time as compatibility effects were relatively similar across the different timepoints (S3 Text). Note, while providing some support, at least some degree of caution is advised surrounding this finding due to the ad hoc nature of the analysis, and introduction of an additional between-measures factor that effectively reduces the number of participants per group.

Despite the plethora of research findings to indicate prime-induced top-down modulation of automatic imitation, which features factors such as social context social context [22, 23, 82], intention [83] and animacy belief [84, 85]), we would be remiss to not consider some of the more recent findings that have contrastingly indicated null or even opposing effects. That is, top-down factors that were originally conceived as being able to module automatic imitation, including social group status [28], prosocial behaviour [86], and belief in free will [87], are beginning to indicate a much smaller or less pervasive influence. In this regard, we may re-

evaluate the capacity for prime-induced top-down modulation of automatic imitation, as well as how this relates to everyday mimicry [88].

While the current stimulus-response paradigm featured procedures designed to avoid any simple spatial compatibility effects (i.e., left-left and right-right stimulus-response mappings), it is possible that some of the present findings may be regarded as complex/orthogonal compatibility effects (i.e., down-left and up-right stimulus-response mappings; [89, 90]) (cf. [55]). However, it is perhaps relevant to note that the present compatibility effects appeared to unfold mostly within the slow rather than fast reaction time bins, although it was not quite corroborated by the Bayes Factor analysis. To elucidate, a compatibility effect unfolding within a slow compared to fast time-course would be more closely associated with automatic imitation, although typical spatial compatibility would unfold regardless of time-course; that is, it would manifest relatively equally in slow and fast reaction time bins [50]. In the absence of a spatial compatibility control task within the present study, we can only reasonably speculate on this possibility. Likewise, there was an even lower number of trials comprising each of the reaction time bins (max. 10 per condition). Future work on automatic imitation should more closely consider the influence of any unintended stimulus-response mappings, while doing so over a potentially more representative number of trials, if indeed compatibility effects are to be categorically attributed to pure automatic imitation.

In conclusion, the present study highlights how situational primes related to the infectious disease, COVID-19, did not modulate the magnitude of automatic imitation in this sample of young UK adults. Importantly, these findings indicate that despite the impact of the COVID-19 pandemic to our daily lives, it does not appear to extend to our natural tendency to imitate others. Future research may further examine the implications of pathogen avoidance and social bonding incentives at times when they are in direct conflict with each other such as during the COVID-19 pandemic. For example, while distancing may appear anti-social within typical circumstances, these are not usual times as distancing may be alternatively seen as prosocial because it offers a level of protection for one's fellow human being. Such lines of inquiry are worthy from a theoretical, but also practical perspective when we consider how the world may be better connected with one another for any future pandemic events.

## Supporting information

**S1 Text. Sentence comprehension task comprising the prime content.**
(DOCX)

**S2 Text. Pre-experimental ratings of valence/pleasantness, arousal, dominance and sociability for the prime content.**
(DOCX)

**S3 Text. Compatibility effect (automatic imitation) as a function of the time of testing.**
(DOCX)

## Author Contributions

**Conceptualization:** James W. Roberts.

**Formal analysis:** James W. Roberts.

**Investigation:** James W. Roberts.

**Methodology:** James W. Roberts, Simon J. Bennett.

**Project administration:** James W. Roberts, Simon J. Bennett.

**Software:** Simon J. Bennett.

**Writing – original draft:** James W. Roberts.

**Writing – review & editing:** Simon J. Bennett.

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
