## [Decision Letter · Decision Letter 0]

17 Aug 2022

PONE-D-22-16580Does the threat of COVID-19 modulate automatic imitation?PLOS ONE

Dear Dr. Roberts,

Thank you for submitting your manuscript to PLOS ONE. After careful consideration, we feel that it has merit but does not fully meet PLOS ONE’s publication criteria as it currently stands. Therefore, we invite you to submit a revised version of the manuscript that addresses the points raised during the review process.

 Can you please address the concerns raised by the reviewers?

We look forward to receiving your revised manuscript.

Kind regards,

Avanti Dey, PhD

Staff Editor

PLOS ONE

Journal Requirements:

2. Please change "female” or "male" to "woman” or "man" as appropriate, when used as a noun (see for instance https://apastyle.apa.org/style-grammar-guidelines/bias-free-language/gender).

Reviewers' comments:

Reviewer's Responses to Questions

**Comments to the Author**

1. Is the manuscript technically sound, and do the data support the conclusions?

Reviewer #1: Partly

Reviewer #2: No

2. Has the statistical analysis been performed appropriately and rigorously? 

Reviewer #1: Yes

Reviewer #2: No

3. Have the authors made all data underlying the findings in their manuscript fully available?

Reviewer #1: Yes

Reviewer #2: Yes

4. Is the manuscript presented in an intelligible fashion and written in standard English?

Reviewer #1: Yes

Reviewer #2: Yes

5. Review Comments to the Author

Reviewer #1: The paper investigates whether the threat of COVID-19 modulates automatic imitation. The concept is interesting though the lack of consideration of existing attitudes towards COVID-19 is problematic.

The participants section should detail the prospective power analyses that were conducted to determine the sample size. If none were conducted then an analysis of the smallest effect size that can be found with the sample size used should be provided. It would also be useful to understand why data collection was initially stopped and then re-started.

More clarity needed in the sentence on lines 101-103 on p12 – unclear what this is referring to.

It would be useful to at least see examples of the sentences used within the method section for context (though I commend you for including the full list within supplementary materials).

Please clarify whether p values are two tailed or one tailed.

Confirmation required on line 156 p 14 as to whether it was the median of the individual participants across the study or the median across participants that was used here.

Perhaps I am misunderstanding but on p8 - 9 you refer to within-participant means and a (lack of a) main effect of prime, but prime was the between subject variable – please clarify this analysis. Given the importance to your hypotheses on the priming factor, I would suggest conducting tests of equivalence or Bayesian statistics to ascertain the likelihood of supportive evidence towards the null in comparison to the alternative hypothesis for this data.

It is questionable whether the interaction on line 208 could be argued to approach conventional levels of significance (particularly if you are reporting one tailed p values). Again, it would be useful to see Bayesian or test of equivalence analyses to support the data (especially given the weight you then put on this in the discussion).

The comparisons of timepoints have small sample sizes, which should be discussed alongside your section on them suggesting there are no differences.

A big omission in the discussion (and in the design of the study) is the pre-existing attitudes towards the pandemic. It has been incredibly divisive with some believing COVID-19 is just like a cold and going about their business as normal. As such, the sentences used are unlikely to change these deep-seeded opinions that COVID-19 is low risk. Having a co-variate of these attitudes would likely have been very informative. Whilst I am not suggesting you conduct another study including this (though if you are conducting further research in this topic I would strongly recommend it), I do think this needs to be discussed as a limitation in the discussion.

Another key limitation that should be discussed is that perhaps one reason for the (lack of) results around the prime is that there was no risk of harm to participants as it was online so no need to adapt their behaviour. Indeed, when online (or when wearing a mask) one may argue that imitation may be even more important in lieu of facial expressions for the latter for example.

Reviewer #2: The manuscript reports findings of an online study on possibility to modulate imitative tendency by contextual primes, specifically those associated with risk of COVID-19 infection. In general, compatibility effect emerged in this dataset and was more prevalent in slower relative to faster response time bins. Tendency to imitate others was not, however, modulated by the situational context related to COVID-19.

After reading the manuscript, my main concern is with its methodology.

First, the main hypothesis seems rather too convoluted to me. I am unsure whether the primes could have been directly relevant for participants in terms of safety or risk. In situation where everyone was bombarded by mass media on COVID-related issues on a daily basis, the participants could have been desensitized to this kind of stimuli. The sample was also relatively young and presumably without serious health issues, so the extent to which they could have been concerned about the ongoing pandemic is questionable. Is there any evidence that these primes had a desired effect (beyond difference in valence, arousal, and dominance)? Many items are related to risk of infection only indirectly and social aspect is not obvious.

Second, the automatic imitation paradigm is not described in sufficient detail and it is therefore unknown to which degree spatial (or other) characteristics might have influenced reported results: (1) the stimulus hand was rotated counter-clockwise, but we do not know whether it was right or left stimulus hand, (2) the participants used right hand to react, but the sample consisted of right- as well as left-handers. The spatial mappings for the latter group are likely to be in opposition to right-handed participants. Moreover, experience in using right hand is not comparable between left- and right-handers, (3) I am not sure whether the stimulus movement was downward (finger press) and response movement upward (finger lift). On page 6, line 123 the authors mention that participants indicated their readiness by pressing the keys, but further down we learn that they also responded by pressing these (already pressed?) keys (line 129). All these aspects could have influenced the magnitude of automatic imitation in this study and they need to be clarified in text and/or illustrated in a figure before it can be established whether the spatial compatibility effects were indeed minimised.

Statistical analyses: 20 trials per condition seem like a rather modest number and I am not sure further division into time bins is meaningful (or necessary – the rationale for this analytical step was not explained). I would also opt for a non-parametric version of ANOVA, given small sample size. Moreover, can we realistically expect significant effects in this small sample? Was the expected effect large enough to be detected in these circumstances (small sample, between-subject design)? At the very least, sensitivity analysis should be conducted to look at the power. ANOVA analyses of standard deviation indices are redundant, I do not understand why they were performed in the first place.

Another issue is the absence of any data concerning personality characteristics (e.g., anxiety). This leaves possible interpretations of a null effect wide open.

Finally, one has to bear in mind that automatic imitation paradigm is essentially a response inhibition task. Social component might not be strong enough and possible spatial influences combined with what I see as weak contextual primes and unknown individual differences of participants could have contributed to reported results.

6. PLOS authors have the option to publish the peer review history of their article (what does this mean?). If published, this will include your full peer review and any attached files.

Reviewer #1: No

Reviewer #2: No

---

## [Author Response · Author response to Decision Letter 0]

29 Sep 2022

*Please note, the references to page and line numbers are taken with respect to the manuscript version, ‘Revised Manuscript with Track Changes’ 

Editor:

Editor’s comment/s: Please change "female” or "male" to "woman” or "man" as appropriate, when used as a noun (see for instance https://apastyle.apa.org/style-grammar-guidelines/bias-free-language/gender).

Authors’ response: We thank the editor for highlighting this oversight. Consequently, we have amended the use of “female” and “male” to appear as “men” and “women” (Pg. 4, Ln. 85).

Editor’s comment/s: We note that you have stated that you will provide repository information for your data at acceptance. Should your manuscript be accepted for publication, we will hold it until you provide the relevant accession numbers or DOIs necessary to access your data. If you wish to make changes to your Data Availability statement, please describe these changes in your cover letter and we will update your Data Availability statement to reflect the information you provide.

Authors’ response: The data are made available through the Open Science Framework (OSF), which has been highlighted within the manuscript (Pg. 8, Ln. 163-164). The assigned DOI is as follows: 10.17605/OSF.IO/3P5DE. We trust this provides sufficient detail to comply with the data availability requirements of the journal, although we can further elaborate if it is deemed necessary.

Editor’s comment/s: Please include captions for your Supporting Information files at the end of your manuscript, and update any in-text citations to match accordingly. Please see our Supporting Information guidelines for more information: https://ddec1-0-en-ctp.trendmicro.com:443/wis/clicktime/v1/query?url=http%3a%2f%2fjournals.plos.org%2fplosone%2fs%2fsupporting%2dinformation&umid=554d1d09-f5a4-4745-a34c-fc46c6b75dbe&auth=6b639a990a359ff1d6cc8761081d57748ce3c81e-3e4f46906a3a111d6721c1f6737a9cead659d5ff.

Authors’ response: Following further revision of our manuscript and the fore mentioned guidelines, we believe the current format complies with the journal requirements including captions pertaining to Supporting Information and in-text citations (i.e., S1-S3 Text).

Reviewers' comments:

Reviewer #1:

Reviewer comment/s: The paper investigates whether the threat of COVID-19 modulates automatic imitation. The concept is interesting though the lack of consideration of existing attitudes towards COVID-19 is problematic.

Authors’ response: We fully recognise the influence of other factors including attitudes, which in the context of the present study, may have mitigated a possible effect on automatic imitation. Consequently, we have expanded upon this issue within the Discussion section, along with some future recommendations for similar research (Pg. 14, Ln. 306-323).

Reviewer comment/s: The participants section should detail the prospective power analyses that were conducted to determine the sample size. If none were conducted then an analysis of the smallest effect size that can be found with the sample size used should be provided. It would also be useful to understand why data collection was initially stopped and then re-started.

Authors’ response: We failed to undertake an apriori power analysis as the sample size was more clearly evaluated with respect to the previous literature featuring a similar research question and design (e.g., Brandt et al., 2017; Cook & Bird, 2011), while we were somewhat limited in the available pool of participants. However, as suggested, we have now incorporated a sensitivity analysis, where the minimal detectable effect size is reported (Pg. 9, Ln. 201-203). Reassuringly, the reported effect sizes that comprised any statistically significant effects appeared greater than the calculated minimal value.

Reviewer’s comment/s: More clarity needed in the sentence on lines 101-103 on p12 – unclear what this is referring to.

Authors’ response: We have attempted to elaborate upon our meaning; that is, we highlight the potential discrepancy between the timeline of events reported within the prime sentences and the calendar month when testing each of the participants (Pg. 5-6, Ln. 110-115).

Reviewer’s comments/: It would be useful to at least see examples of the sentences used within the method section for context (though I commend you for including the full list within supplementary materials).

Authors’ response: We have incorporated some examples for each of the prime categories (i.e., control, safe, unsafe) with a view to highlighting their true nature primarily within the Methods section (Pg. 6, Ln. 117-120), and also within the Discussion section (Pg. 13, Ln. 281 and 288).

Reviewer’s comment/s: Please clarify whether p values are two tailed or one tailed.

Authors’ response: We have indicated the direction of the probability distribution pertaining to the p-value from the inferential statistics; that is, it was two-tailed (Pg. 8, Ln. 182).

Reviewer’s comment/s: Confirmation required on line 156 p 14 as to whether it was the median of the individual participants across the study or the median across participants that was used here.

Authors’ response: We have attempted to clarify how this particular data were handled; that is, we split the data on a participant-by-participant basis, where the median was taken for each individual participant (Pg. 8, Ln. 170-171).

Reviewer’s comment/s: Perhaps I am misunderstanding but on p8-9 you refer to within-participant means and a (lack of a) main effect of prime, but prime was the between subject variable – please clarify this analysis. Given the importance to your hypotheses on the priming factor, I would suggest conducting tests of equivalence or Bayesian statistics to ascertain the likelihood of supportive evidence towards the null in comparison to the alternative hypothesis for this data.

Authors’ response: We would like to thank the reviewer for suggesting the fore mentioned analysis, which was overlooked in our previous version of the manuscript. Consequently, we undertook a series of Bayesian ANOVAs (Pg. 8-9, Ln. 184-191), and have now incorporated the Bayes factor (BF10; <1 indicating less support for the alternative over the null hypothesis) into each one of the statistical effects/models (Pg. 9-10; see also, S3 Text). What’s more, we have adapted this detail by incorporating it into the Discussion section as part of our central or main interpretation of the findings (Pg. 12, Ln. 274-275).

Reviewer’s comment/s: It is questionable whether the interaction on line 208 could be argued to approach conventional levels of significance (particularly if you are reporting one tailed p values). Again, it would be useful to see Bayesian or test of equivalence analyses to support the data (especially given the weight you then put on this in the discussion).

Authors’ response: Please note, the fore mentioned statistical effect pertains to the within-participant standard deviation in reaction time, which in line with another reviewer’s comments, has now been removed as a dependent variable. Consequently, the fore mentioned line number and its content no longer exist.

With regard the Bayesian analysis, please see the adjacent response regarding Bayesian ANOVAs.

Reviewer’s comment/s: The comparisons of timepoints have small sample sizes, which should be discussed alongside your section on them suggesting there are no differences.

Authors’ response: We fully acknowledge the small group sizes comprising our ad hoc between-measures factor of timepoint. Consequently, we advise the reader not to form any definitive conclusions based on this particular analysis (Pg. 15, Ln. 333-336).

Reviewer’s comment/s: A big omission in the discussion (and in the design of the study) is the pre-existing attitudes towards the pandemic. It has been incredibly divisive with some believing COVID-19 is just like a cold and going about their business as normal. As such, the sentences used are unlikely to change these deep-seeded opinions that COVID-19 is low risk. Having a co-variate of these attitudes would likely have been very informative. Whilst I am not suggesting you conduct another study including this (though if you are conducting further research in this topic I would strongly recommend it), I do think this needs to be discussed as a limitation in the discussion.

Authors’ response: We fully recognise the issue of other confounding variables or factors that may have mitigated an effect of the primes on automatic imitation. In this regard, we have attempted to address at least some of these key factors, and further still, how they ought to be considered for similar future research (Pg. 14, Ln. 306-323).

Reviewer’s comment/s: Another key limitation that should be discussed is that perhaps one reason for the (lack of) results around the prime is that there was no risk of harm to participants as it was online so no need to adapt their behaviour. Indeed, when online (or when wearing a mask) one may argue that imitation may be even more important in lieu of facial expressions for the latter for example.

Authors’ response: While online, it is important to stress that the current study featured similar task procedures as previous other studies that have demonstrated prime-induced top-down modulation (Pg. 13, Ln. 282-285). In this regard, we deem it to be reasonable to explore the possibility of situational primes to infectious disease on mimicry-type behaviour using the current computer-based stimulus-response paradigm.

Reviewer #2:

Reviewer comment/s: The manuscript reports findings of an online study on possibility to modulate imitative tendency by contextual primes, specifically those associated with risk of COVID-19 infection. In general, compatibility effect emerged in this dataset and was more prevalent in slower relative to faster response time bins. Tendency to imitate others was not, however, modulated by the situational context related to COVID-19.

After reading the manuscript, my main concern is with its methodology.

First, the main hypothesis seems rather too convoluted to me. I am unsure whether the primes could have been directly relevant for participants in terms of safety or risk. In situation where everyone was bombarded by mass media on COVID-related issues on a daily basis, the participants could have been desensitized to this kind of stimuli. The sample was also relatively young and presumably without serious health issues, so the extent to which they could have been concerned about the ongoing pandemic is questionable. Is there any evidence that these primes had a desired effect (beyond difference in valence, arousal, and dominance)? Many items are related to risk of infection only indirectly and social aspect is not obvious.

Authors’ response: We have adapted our hypothesis; namely, we have attempted to expand upon the rationale or reasoning behind our predicted direction of effects (Pg. 4, Ln. 73-79).

Second, the automatic imitation paradigm is not described in sufficient detail and it is therefore unknown to which degree spatial (or other) characteristics might have influenced reported results: (1) the stimulus hand was rotated counter-clockwise, but we do not know whether it was right or left stimulus hand, (2) the participants used right hand to react, but the sample consisted of right- as well as left-handers. The spatial mappings for the latter group are likely to be in opposition to right-handed participants. Moreover, experience in using right hand is not comparable between left- and right-handers, (3) I am not sure whether the stimulus movement was downward (finger press) and response movement upward (finger lift). On page 6, line 123 the authors mention that participants indicated their readiness by pressing the keys, but further down we learn that they also responded by pressing these (already pressed?) keys (line 129). All these aspects could have influenced the magnitude of automatic imitation in this study and they need to be clarified in text and/or illustrated in a figure before it can be established whether the spatial compatibility effects were indeed minimised.

Authors’ response: We have adapted the methodology including matters of handedness (Pg. 6, Ln. 131) and response key presses/releases (Pg. 7, Ln. 137-139). Please note, if the present changes remain inadequate, then we would welcome the design of a figure in an attempt to alternatively illustrate the paradigm.

Moreover, all participants were exposed to only a left-hand stimulus and required to respond using only their right hand, which meant the stimulus-response mappings were spatially the same for all participants. The failure to categorise participants as a function of handedness ensured greater inclusivity, as well as a larger and more meaningful sample. However, for further reassurance, inspection of the automatic imitation effects for the right- and left-handed participants revealed that they were somewhat comparable (right-handed M diff ± SD = 42 ms ± 72; left-handed M diff ± SD = 49 ms ± 57).

Reviewer’s comment/s: 20 trials per condition seem like a rather modest number and I am not sure further division into time bins is meaningful (or necessary – the rationale for this analytical step was not explained).

Authors’ response: The selected number of trials per condition is aligned with the range adopted in previous other studies that have similarly demonstrated an automatic imitation effect (e.g., Leighton et al., 2010; Spengler et al., 2010; Wang & Hamilton, 2013). Indeed, the pervasive nature of the automatic imitation effect (Cracco et al., 2018) means there are comparatively few constraints that limit its occurrence including the number of trials. Additionally, it is important to note that participants were granted initial practice at the task such that the 20 reported experimental trials of interest could be fully representative of the participants’ reaction times in each condition. Finally, from a pragmatic perspective, we felt the current selection of trials was adequate enough to both capture automatic imitation, while ensuring adherence to the task procedures despite the remote online nature of the data collection in the absence of any experimenter oversight.

With regard the analysis that incorporates the factor of time bin, we adapted this from previous other studies that have taken a similar approach to examining automatic imitation (e.g.. Catmur & Heyes, 2010; Cook & Bird, 2011). Indeed, it provides further detail surrounding the nature or direction of automatic imitation – does it manifest from the facilitation or inhibition of imitative responses? We have already attempted to provide an explanation or rationale for this particular analysis (Pg. 8, Ln. 172-175), although we welcome further suggestions to elaborate if indeed it is deemed necessary.

Reviewer’s comment/s: I would also opt for a non-parametric version of ANOVA, given small sample size. Moreover, can we realistically expect significant effects in this small sample? Was the expected effect large enough to be detected in these circumstances (small sample, between-subject design)? At the very least, sensitivity analysis should be conducted to look at the power.

Authors’ response: We are unaware of any information to suggest that the current sample size was too small within the context of the present study on automatic imitation. Indeed, the current sample size is either aligned or exceeds the majority of empirical research studies involving automatic imitation (e.g., Brandt et al., 2017; Cook & Bird, 2011). This logic is supported by the notion that the automatic imitation effect is sufficiently powerful and pervasive that it manifests consistently in neurotypical individuals (Cracco et al., 2018). That said, we fully acknowledge the absence of an apriori power analysis that would have otherwise informed us of the necessary sample size. Thus, in line with the fore mentioned suggestion, we have conducted a sensitivity analysis with a view to calculating the minimal detectable effect size to be considered significant or true (Pg. 9, Ln. 201-203). The outcome of this analysis confirmed that the existing statistically significant effects featured effect sizes that were greater than the minimal detectable value. In addition, any remaining null effects coincided with sufficiently low effect sizes.

Reviewer’s comment/s: ANOVA analyses of standard deviation indices are redundant, I do not understand why they were performed in the first place.

Authors’ response: We fully acknowledge the use of within-participant standard deviation as being somewhat redundant; particularly as it garners a limited interpretation or subsequent discussion. As a result, we have removed this particular dependent measure, which means only the participant mean reaction times are used as the main dependent measure.

Reviewer’s comment/s: Another issue is the absence of any data concerning personality characteristics (e.g., anxiety). This leaves possible interpretations of a null effect wide open.

Finally, one has to bear in mind that automatic imitation paradigm is essentially a response inhibition task. Social component might not be strong enough and possible spatial influences combined with what I see as weak contextual primes and unknown individual differences of participants could have contributed to reported results.

Authors’ response: We recognise the limitation surrounding the absence of individual differences that may factor into the interpretation and subsequent influence of the situational primes. With this in mind, we have elaborated upon this limitation including future research recommendations within the Discussion section (Pg. 14, Ln. 306-323).

With regard the sensitivity to contextual priors within the present paradigm, it is important to note that numerous other studies have positively evidenced top-down social modulation of automatic imitation within a similar computer-based stimulus-response paradigm (Pg 13., Ln. 282-285). In this regard, the present study is heavily adapted from what is perhaps considered a well-established or pervasive empirical research finding, where the social affiliative behaviour that is related to daily mimicry can be successfully captured within the present paradigm.

References:

Brandt VC, Moczydlowski A, Jonas M, Boelmans K, Bäumer T, Brass M, et al. Imitation inhibition in children with Tourette syndrome. J Neuropsychol, 2019;13(1):82-95.

Catmur C, Heyes C. Time course analyses confirm independence of imitative and spatial compatibility. J Exp Psychol Hum Percept Perform. 2011;37(2):409-21.Cook & Bird, 2011

Cracco E, Bardi L, Desmet C, Genschow O, Rigoni D, De Coster L, et al. Automatic imitation: a meta-analysis. Psychol Bull. 2018;144(5):453-500.

Leighton J, Bird G, Orsini C, Heyes C. Social attitudes modulate automatic imitation. J Exp Soc Psychol. 2010; 46(6):905-10.

Spengler S, Brass M, Kühn S, Schütz-Bosbach S. Minimizing motor mimicry by myself: self-focus enhances online action-control mechanisms during motor contagion. Conscious Cogn. 2010;19(1):98-106.

Wang Y, Hamilton AFC. Understanding the role of the ‘self’ in the social priming of mimicry. PLoS ONE. 2013; 8(4):e60249.

---

## [Decision Letter · Decision Letter 1]

19 Oct 2022

PONE-D-22-16580R1Does the threat of COVID-19 modulate automatic imitation?PLOS ONE

Dear Dr. Roberts,

Thank you for submitting your manuscript to PLOS ONE. After careful consideration, we feel that it has merit but does not fully meet PLOS ONE’s publication criteria as it currently stands. Therefore, we invite you to submit a revised version of the manuscript that addresses the points raised during the review process.

As you will see, reviewer 2 is still conerned about the issue of spatial compatibility between the stimulus and the response, which is ususally overcome by using orthogonal orientations between the direction of movement and the observer's or imitator's personal body part. It is not proposed that you run ocntrol experiments, but it is requested that you explicitely and extensivley acknowledge the fact that the effect you are measuring is spurious and contains an un-measurable proportion of simple spatial compatibility rather than genuine action observation effect. This is actually a very big issue in action observation protocols, whcih should always be controlled fro spatial compatibility issues.

We look forward to receiving your revised manuscript.

Kind regards,

Luigi Cattaneo, MD, PhD

Academic Editor

PLOS ONE

Additional Editor Comments (if provided):

As you will see, reviewer 2 is still conerned about the issue of spatial compatibility between the stimulus and the response, which is ususally overcome by using orthogonal orientations between the direction of miovement and the observer's or imitator's personal body part. It is not proposed that you run ocntrol experiments, but it is requested that you explicitely and extensivley acknowledge the fact that the effect you are measuring is spurious and contains an un-measurable proportion of simple spatial compatibility rather compared to the genuine action observation effect. This is actually a very big issue in action observation protocols, whcih should always be controlled fro spatial compatibility issues.

Reviewers' comments:

Reviewer's Responses to Questions

**Comments to the Author**

1. If the authors have adequately addressed your comments raised in a previous round of review and you feel that this manuscript is now acceptable for publication, you may indicate that here to bypass the “Comments to the Author” section, enter your conflict of interest statement in the “Confidential to Editor” section, and submit your "Accept" recommendation.

Reviewer #1: All comments have been addressed

Reviewer #2: (No Response)

2. Is the manuscript technically sound, and do the data support the conclusions?

Reviewer #1: Yes

Reviewer #2: No

3. Has the statistical analysis been performed appropriately and rigorously? 

Reviewer #1: Yes

Reviewer #2: No

4. Have the authors made all data underlying the findings in their manuscript fully available?

Reviewer #1: Yes

Reviewer #2: Yes

5. Is the manuscript presented in an intelligible fashion and written in standard English?

Reviewer #1: Yes

Reviewer #2: Yes

6. Review Comments to the Author

Reviewer #1: Thank you for your responses. It is surprising that apriori power analyses weren't conducted. In the participants section you should detail what the current sample size was based on.

Reviewer #2: After reading the revised version of the MS, my concerns over methodology of this study remain.

On the basis of paradigm description provided in the revised MS I believe that orthogonal-spatial compatibility effects were not considered and the measured effect confounds spatial and imitative influences. Although many existing studies do not take spatial effects into account in their designs (either simple or orthogonal), sufficient evidence on this issue is now available (see papers on automatic imitation paradigm as well as other spatial compatibility tasks). Second, these spatial compatibility effects differ for right- and left-handers. While both subgroups might have had a similar magnitude of compatibility effect in this particular sample, its primary sources can be different for each subgroup. The resulting effect could be due predominantly to orthogonal-spatial compatibility in the right-handers, while the opposite could be true for left-handers. In other words, while inclusion of left-handers ensured larger sample, the measured effects became more confounded. In this context, it is irrelevant that the same stimulus was presented to the entire sample – all responses were confounded to some extent (and this extent is unknown). Given that additional data collection to address this methodological oversight is not feasible at this stage, the issue of orthogonal compatibility has to be reflected in interpretation and discussion of the results because a lack of behavioural modification by social cues can well be due to measurement of spatial, not imitative characteristics. I strongly suggest that authors become familiar with issues associated with automatic imitation paradigm – specifically those related to spatial confounds and external validity of the paradigm (e.g., recently reported lack of association between imitation inhibition using similar paradigm and mimicry).

Rationale for modest number of trials (i.e., to ensure cooperation of participants) should be provided in the Method section. I disagree with the claim that majority of studies on automatic imitation comprise similar number of trials/participants – large-scale reports are also available. Moreover, individual differences in automatic imitation effect have been revealed and automatic imitation does not manifest consistently at individual level even in neurotypical participants, nor it has a “standard level” (page 4, line 76).

Minor:

- How did the authors arrive at the reported minimum effect size (ηp2= .043)? Please specify.

- It would be useful to report statistical indices including effect size for contrasts in case of a significant interaction term.

- Sentences on page 15, lines 330-336 belong to the Result section.

In conclusion, my main concern is that it is impossible to tell what was measured in the present study. It is likely that the version of automatic imitation paradigm captures non-social (spatial) rather than social (imitative) compatibility effects. A revision of the MS, where the authors acknowledge this issue and incorporate orthogonal compatibility effect into their interpretations.

7. PLOS authors have the option to publish the peer review history of their article (what does this mean?). If published, this will include your full peer review and any attached files.

Reviewer #1: No

Reviewer #2: No

---

## [Author Response · Author response to Decision Letter 1]

24 Oct 2022

*Please note, the references to page and line numbers are taken with respect to the manuscript version, ‘Revised Manuscript with Track Changes’ 

Editor:

Editor comment/s: As you will see, reviewer 2 is still concerned about the issue of spatial compatibility between the stimulus and the response, which is usually overcome by using orthogonal orientations between the direction of movement and the observer's or imitator's personal body part. It is not proposed that you run control experiments, but it is requested that you explicitly and extensively acknowledge the fact that the effect you are measuring is spurious and contains an un-measurable proportion of simple spatial compatibility rather compared to the genuine action observation effect. This is actually a very big issue in action observation protocols, which should always be controlled for spatial compatibility issues.

Authors’ response: We fully recognise the potential of spatial compatibility effects contaminating the present findings (Pg. 14, Ln. 319-330). Specifically, we highlight the potential for complex/orthogonal spatial compatibility, where the left-down and right-up stimulus-response mappings may be observed (N.B., only for midline or right hemispace responses).

Reviewers' comments:

Reviewer #1:

Reviewer comment/s: Thank you for your responses. It is surprising that apriori power analyses weren't conducted. In the participants section you should detail what the current sample size was based on.

Authors’ response: We would like to thank the reviewer for their constructive comments. The sample size in question was mostly determined by the comparatively limited uptake when recruiting participants combined with the knowledge that the pandemic and national restrictions were nearing an end (Pg. 4-5, Ln. 86-87). That said, the subsequent sample size was consistent with much of the previous literature (Leighton et al., 2010; Wang & Hamilton, 2013) (Pg. 5, Ln. 87-88), while the sensitivity analysis was favourable toward our obtaining an achievable effect size (Pg. 9, Ln. 194-196). In this regard, we consider our sample size sufficient enough for the purposes of the present study.

Reviewer #2:

Reviewer’s comment/s: On the basis of paradigm description provided in the revised MS I believe that orthogonal-spatial compatibility effects were not considered and the measured effect confounds spatial and imitative influences. Although many existing studies do not take spatial effects into account in their designs (either simple or orthogonal), sufficient evidence on this issue is now available (see papers on automatic imitation paradigm as well as other spatial compatibility tasks). Second, these spatial compatibility effects differ for right- and left-handers. While both subgroups might have had a similar magnitude of compatibility effect in this particular sample, its primary sources can be different for each subgroup. The resulting effect could be due predominantly to orthogonal-spatial compatibility in the right-handers, while the opposite could be true for left-handers. In other words, while inclusion of left-handers ensured larger sample, the measured effects became more confounded. In this context, it is irrelevant that the same stimulus was presented to the entire sample – all responses were confounded to some extent (and this extent is unknown). Given that additional data collection to address this methodological oversight is not feasible at this stage, the issue of orthogonal compatibility has to be reflected in interpretation and discussion of the results because a lack of behavioural modification by social cues can well be due to measurement of spatial, not imitative characteristics. I strongly suggest that authors become familiar with issues associated with automatic imitation paradigm – specifically those related to spatial confounds and external validity of the paradigm (e.g., recently reported lack of association between imitation inhibition using similar paradigm and mimicry).

Authors’ response: We fully recognise the potential of spatial compatibility effects contaminating the present findings (Pg. 14, Ln. 319-323). Specifically, we highlight the potential for complex/orthogonal spatial compatibility, where the left-down and right-up stimulus-response mappings may be observed (N.B., only for midline or right hemispace responses).

However, we are unable to reconcile the influence of handedness on such spatial compatibility effects. This is not to say that such an influence does not exist, although we currently fail to comprehend this logic (for response hand effects, see Cho & Proctor, 2004). Along these lines, we may indirectly infer the extent of spatial compatibility within right- and left-handed participants by re-evaluating the compatibility effects as a function of time bin (i.e., slow vs. fast). Here, it would be suggested that compatibility effects primarily unfolding within slow compared to fast reaction time bins would perhaps be more closely associated with automatic imitation (Catmur & Heyes, 2011) (Pg. 14, Ln. 323-327). Meanwhile, compatibility effects that unfold relatively equally across time bins would perhaps earmark spatial compatibility. With this in mind, we found a similarly increased compatibility effect within the slow compared to fast time bins for right- (slow M diff ± SD =53 ms ± 104; fast M diff ± SD = 34 ms ± 59) and left-handed (slow M diff ± SD =50 ms ± 63; fast M diff ± SD = 36 ms ± 59) participants 

Reviewer’s comment/s: Rationale for modest number of trials (i.e., to ensure cooperation of participants) should be provided in the Method section. I disagree with the claim that majority of studies on automatic imitation comprise similar number of trials/participants – large-scale reports are also available.

Authors’ response: We fully agree that while the currently adopted number of trials is broadly consistent with those adopted within previous studies (e.g., Wang & Hamilton, 2013), it is not consistent with a large amount of other studies surrounding this area of work. Consequently, and as requested, we have provided further detail on our choice of the number of trials (Pg. 7, Ln. 149-152).

Reviewer’s comment/s: Moreover, individual differences in automatic imitation effect have been revealed and automatic imitation does not manifest consistently at individual level even in neurotypical participants, nor does it have a “standard level” (page 4, line 76).

Authors’ response: We fully recognise the emerging consensus surrounding the heterogeneity of automatic imitation within the Discussion section (Pg. 14, Ln. 310-318), and alternatively adopt the terminology “typical or baseline” (i.e., within-participant) as opposed to “standard level” (Pg. 4, Ln. 71).

Reviewer’s comment/s: How did the authors arrive at the reported minimum effect size (ηp2= .043)? Please specify.

Authors’ response: In line with another reviewer’s request, the minimum effect size to which the test was sensitive was calculated using G*Power. The precise inputs were as follows: Test Family > F Test; Statistical Test > ANOVA – Repeated-measures, within-between; Type of power analysis > Sensitivity; α = .05; 1-β = .80; Total sample = 59; Number of groups = 3; Number of measurements = 2. The relevant parameters for this sensitivity test are stated within the manuscript (Pg. 9, Ln. 194-196).

Reviewer’s comment/s: It would be useful to report statistical indices including effect size for contrasts in case of a significant interaction term.

Authors’ response: We are not entirely certain here of the request alluding to effect sizes in addition to the ones currently present for each inferential statistic. Naturally, we welcome further consideration upon potentially receiving another iteration of reviews.

Reviewer’s comment/s: Sentences on page 15, lines 330-336 belong to the Results section.

Authors’ response: We fully recognise the principle of describing findings initially within the Results section prior to the Discussion section. However, the findings of interest here are related to an ad hoc observation that pertains to a critical evaluation of the study as it nears its completion. Thus, in order to relocate the description of these particular findings to the Results section, then they would need to be presented prior to forming a rationale in terms of why this analysis was conducted in the first place. Moreover, there are some─all be it rare─instances where similar ad hoc observations are formulated within the Discussion section (e.g., Buckingham et al., 2014; Spengler et al., 2010; Wong et al., 2013). At the same time, it is worthwhile highlighting that the associated inferential statistics are presented elsewhere as a supplement (S3 Text). With this in mind, we have currently elected to retain the description of the findings in question within the Discussion section. Naturally, we welcome the need to revisit this choice if indeed our current explanation is not deemed sufficient.

Reviewers’ comment/s: In conclusion, my main concern is that it is impossible to tell what was measured in the present study. It is likely that the version of automatic imitation paradigm captures non-social (spatial) rather than social (imitative) compatibility effects. A revision of the MS, where the authors acknowledge this issue and incorporate orthogonal compatibility effect into their interpretations.

Authors’ response: We fully recognise this possibility within the Discussion section (Pg. 14, Ln. 319-323). Please see previous response to comment/s regarding complex/orthogonal spatial compatibility.

References

Buckingham G, Wong JD, Tang M, Gribble PL, Goodale MA. Observing object lifting errors modulates cortico-spinal excitability and improves object lifting performance. Cortex. 2014; 50:115-24.

Catmur C, Heyes C. Time course analyses confirm independence of imitative and spatial compatibility. J Exp Psychol Hum Percept Perform. 2011; 37(2):409-21.

Cho YS, Proctor RW. Influences of multiple spatial stimulus and response codes on orthogonal stimulus–response compatibility. Percept Psychophys. 2004; 66:1003-17.

Leighton J, Bird G, Orsini C, Heyes C. Social attitudes modulate automatic imitation. J Exp Soc Psychol. 2010; 46(6):905-10.

Spengler S, von Cramon DY, Brass M. Resisting motor mimicry: control of imitation involves processes central to social cognition in patients with frontal and temporo-parietal lesions. Soc Neurosci. 2010; 5(4):401-16.

Wang Y, Hamilton AFC. Understanding the role of the ‘self’ in the social priming of mimicry. PLoS ONE. 2013; 8(4):e60249.

Wong L, Manson GA, Tremblay L, Welsh TN. On the relationship between the execution, perception, and imagination of action. Behav Brain Res. 2013; 257:242-52.

---

## [Decision Letter · Decision Letter 2]

23 Jan 2023

PONE-D-22-16580R2Does the threat of COVID-19 modulate automatic imitation?PLOS ONE

Dear Dr. Roberts,

Thank you for submitting your manuscript to PLOS ONE. After careful consideration, we feel that it has merit but does not fully meet PLOS ONE’s publication criteria 3 and 4 as it currently stands.

As you will see below, the newly included reviewer highlighted relevant points regarding the inclusion of current literature, the rationale for the design as well as other issues with the analyses and reporting thereof. We feel that addressing these points as outlined in their review will strengthen your manuscript.

You will also note that whilst both reviewers provided relevant suggestions on how to address issues of compatibility, their reading of the analysis on both compatibility and time bin interaction differs. I understand that including both factors (compatibility and time-bin interaction) would require a hierarchical analysis approach. To complicate matters further, I have another suggestion by using a different kind of analysis altogether. I would have expected to see a linear mixed-effects model analysis for this type of data/research question. This would have prevented some of the issues identified by the reviewers while at the same time considering other factors (timepoint, gender, differences in priming sentences, individual participant data; and thus more powerful). 

My concern regarding the bin-analysis in its current form is that a higher variance in longer time intervals is simply more likely due to longer time intervals. Please clarify this point. Also, a rationale regarding the fast/slow response mode is missing in the introduction. Was this a post-hoc exploratory analysis? If so, this needs to be stated.

Therefore, we invite you to submit a revised version of the manuscript that addresses all the points raised during this reviewing cycle. As for the issues surrounding your compatibility analysis, please address all comments and requests for additional data and presentation of findings and clarify the analysis as well as the rationale in the manuscript and/or the response letter.

In addition to the revision, I urge you to include the individual trial RT data in your OSF repository plus participants’ gender (or sex, depending how the participants demographics question). For the data to be of value for certain types of analyses, individual trial data is needed, and gender seems relevant, considering that males have been found to be less Covid-19 risk adverse than females (and you should in fact consider the unbalanced participant cohort as one of your limitations). Finally, I agree with the reviewer for the need of a figure illustrating the stimuli/task in the manuscript and I would recommend uploading your visuals onto the repository.

We look forward to receiving your revised manuscript.

Kind regards,

Corinne Jola

Academic Editor

PLOS ONE

Journal Requirements:

Additional Editor Comments (if provided):

We invite you to submit a revised version of the manuscript that addresses all the points raised during this reviewing cycle. As for the issues surrounding your compatibility analysis, please address all comments and requests for additional data and presentation of findings and clarify the analysis as well as the rationale in the manuscript and/or the response letter. Please see further comments in my response letter as well as the reviewers' comments.

Reviewers' comments:

Reviewer's Responses to Questions

**Comments to the Author**

1. If the authors have adequately addressed your comments raised in a previous round of review and you feel that this manuscript is now acceptable for publication, you may indicate that here to bypass the “Comments to the Author” section, enter your conflict of interest statement in the “Confidential to Editor” section, and submit your "Accept" recommendation.

Reviewer #2: (No Response)

Reviewer #3: (No Response)

2. Is the manuscript technically sound, and do the data support the conclusions?

Reviewer #2: Partly

Reviewer #3: Partly

3. Has the statistical analysis been performed appropriately and rigorously? 

Reviewer #2: Yes

Reviewer #3: Yes

4. Have the authors made all data underlying the findings in their manuscript fully available?

Reviewer #2: Yes

Reviewer #3: Yes

5. Is the manuscript presented in an intelligible fashion and written in standard English?

Reviewer #2: Yes

Reviewer #3: Yes

6. Review Comments to the Author

Reviewer #2: Although the authors added a brief section on spatial compatibility effects in the revised MS, they support their main interpretation using papers building on different paradigms with potentially different spatial relationships (e.g., participants using both hands). I am therefore unsure if these results can be generalized to the presented study. It has already been revealed that the stimulus hand used here elicits predominantly orthogonal compatibility effects and that purely imitative effects exist, but are rare, more modest in magnitude, and likely elicited by different stimulus. Other stimuli/contexts might not elicit significant compatibility effects at all. This is the reason why reporting strength of the effect (incom-com contrasts) in slow and fast bins is important (compatibility x time bin interaction). Were all incom-com differences in slow and fast bins significant? This could help the interpretation. For example, numerical difference in fast bin between incom and com trials for neutral condition seems very small (12ms) compared to the same difference in unsafe condition (49ms). Moreover, it would be easier on a reader to present these results in a graph rather than table.

Reviewer #3: In this manuscript the authors used an automatic imitation task preceded by a sentence comprehension priming task to test for the influence of COVID-related primes on imitation. An imitation effect was observed but no influence of the priming task.

I appreciate that this manuscript has already been reviewed and it’s galling to have to deal with a new set of comments after two rounds of review, so I leave it to the editor’s discretion to decide which of the following points are essential for the authors to address.

Introduction

The focus on ‘advanced mammals’ in the first sentence seems unnecessary and inaccurate – a wide range of species show forms of imitation, including birds (Richards et al., 2009).

The introduction is missing discussion of studies which do not show an effect of social priming on imitation, including several recent and more highly powered studies (Genschow et al., 2022; see also the recent special issue on this topic by Cracco et al., 2022).

The rationale for the study doesn’t follow from the discussion of the BIS in the introduction. According to the introduction, the effects of the COVID-related primes should only be seen in participants with tendencies to implement avoidance behaviours (last paragraph of page 3) making it unclear why any effect of COVID-related primes on imitation should be found in a more general population sample?

Method

Please provide a clearer statement of the statistical power the sample size afforded. The two studies cited here are a decade old and more recent studies have used far larger sample sizes (again see Cracco et al. special issue / editorial). Although you can’t estimate the likely size of the effect you are trying to detect, you can at least state what effect you were powered to detect (e.g. ‘a sample of 20 participants per group provides 80% power to detect an effect of at least d=0.9 at an alpha level of .05’).

Please provide a figure illustrating the stimuli/task.

20 trials per condition is a lot fewer than in most automatic imitation tasks. Please provide rationale in terms of the number of trials needed to obtain a robust estimate of response time in each condition, and also indicate the mean/SD number of trials per condition remaining once false starts, delayed responses, and error trials were removed; and discuss how this relates to the number of trials required for estimating RT.

Results

Given the very low number of trials I don’t think the bin analysis is appropriate. At the very least the authors should report the mean/SD number of trials remaining within each bin.

It would be helpful to add the magnitude of the compatibility effects to Table 1 (i.e. the mean/SE of the Incompatible-Compatible difference across participants), please.

Last paragraph of page 9 – this states that the dependent variable was the compatibility effect ‘Further inspection of the compatibility effects as a function of movement time bins’ but then compatibility is reported as a factor in the analysis – so is this actually analysing RT not compatibility effect? Please clarify.

First lines of page 10 – the Bayes’ factor associated with this interaction is in favour of the null hypothesis, suggesting limited support for different magnitude compatibility effects across the slow and fast bins – please check and then reflect this in your discussion (if indeed, given my points about trial numbers above, you retain the bin analysis).

Discussion

Page 14 – ‘the present compatibility effects unfolded within the slow rather than fast reaction time bins’ – this isn’t quite accurate, the effects were greater in the slow bins but they were still present in the fast bins? A follow-up analysis of the simple effects of compatibility at each time bin would help to address this point.

I think the conclusion should be qualified ‘did not modulate automatic imitation in this group of young UK adults’.

Richards C, Mottley K, Pearce J, Heyes C. 2009. Imitative pecking by budgerigars, Melopsittacus undulatus, over a 24 h delay. Anim. Behav. 77, 1111–1118. ( 10.1016/j.anbehav.2009.01.019)

Genschow, O., Westfal, M., Cracco, E., & Crusius, J. (2022). Group membership does not modulate automatic imitation. Psychological research, 86(3), 780–791. https://doi.org/10.1007/s00426-021-01526-1

Cracco, E., Genschow, O., & Baess, P. (2022). Top-down social modulation of perception-action coupling. Acta psychologica, 222, 103481. https://doi.org/10.1016/j.actpsy.2021.103481

7. PLOS authors have the option to publish the peer review history of their article (what does this mean?). If published, this will include your full peer review and any attached files.

Reviewer #2: No

Reviewer #3: No

---

## [Author Response · Author response to Decision Letter 2]

9 Mar 2023

Editor:

Editor comment/s: As you will see below, the newly included reviewer highlighted relevant points regarding the inclusion of current literature, the rationale for the design as well as other issues with the analyses and reporting thereof. We feel that addressing these points as outlined in their review will strengthen your manuscript.

Reviewer’s comment/s: You will also note that whilst both reviewers provided relevant suggestions on how to address issues of compatibility, their reading of the analysis on both compatibility and time bin interaction differs. I understand that including both factors (compatibility and time-bin interaction) would require a hierarchical analysis approach. To complicate matters further, I have another suggestion by using a different kind of analysis altogether. I would have expected to see a linear mixed-effects model analysis for this type of data/research question. This would have prevented some of the issues identified by the reviewers while at the same time considering other factors (timepoint, gender, differences in priming sentences, individual participant data; and thus more powerful). 

Authors’ response: We are not explicitly clear on the need to adopt a linear mixed effects model within the present study, which otherwise features a fully balanced study design. Most importantly, and as alluded to within the comments above, we potentially risk compromising the requested revisions pertaining to a separate set of reviewer comments, which otherwise continue to endorse a general linear model. With this in mind, we have alternatively attempted to thoroughly acknowledge and/or remedy the above raised issues within the present set of revisions (see below).

My concern regarding the bin-analysis in its current form is that a higher variance in longer time intervals is simply more likely due to longer time intervals. Please clarify this point. Also, a rationale regarding the fast/slow response mode is missing in the introduction. Was this a post-hoc exploratory analysis? If so, this needs to be stated.

Authors’ response: We would like to thank the editor for this observation. As a result, we fully recognise the possible confounding influence of variance, where the magnitude of differences could be coincidentally scaled to the magnitude of time; thus, indicating heteroscedasticity. That said, we would perhaps assume that any such influence would render differences between compatible and incompatible trials that are completely random, and not systematic as found within the present study (i.e., compatibility effect-mean reaction time relation would effectively resemble a ‘funnel’ and not necessarily a systematic linear relation).

For context, the original logic behind the use of time bins assumed that a compatibility effect related to biologically-specific stimuli, which is synonymous with automatic imitation, would be mostly prevalent within slower time bins. On the other hand, a simple compatibility effect featuring arbitrary, non-biological stimuli would unfold relatively equally across time bins (Catmur & Heyes, 2011). Nevertheless, in the absence of a spatial compatibility control task within the present study, we concede that it is not suitable to positively assert this logic (Pg. 15-16, Ln. 355-357).

Of interest, the analysis involving time bins was based on an apriori rationale following previous empirical research findings (Catmur & Heyes, 2011; Cook & Bird, 2011; Heyes & Bird, 2007; Spengler et al., 2010), and thus we have more precisely reported it within the Introduction section including an hypothesis (Pg. 4, Ln. 77-86).

Therefore, we invite you to submit a revised version of the manuscript that addresses all the points raised during this reviewing cycle. As for the issues surrounding your compatibility analysis, please address all comments and requests for additional data and presentation of findings and clarify the analysis as well as the rationale in the manuscript and/or the response letter.

In addition to the revision, I urge you to include the individual trial RT data in your OSF repository plus participants’ gender (or sex, depending how the participants demographics question). For the data to be of value for certain types of analyses, individual trial data is needed, and gender seems relevant, considering that males have been found to be less Covid-19 risk adverse than females (and you should in fact consider the unbalanced participant cohort as one of your limitations). Finally, I agree with the reviewer for the need of a figure illustrating the stimuli/task in the manuscript and I would recommend uploading your visuals onto the repository.

Authors’ response: We have now incorporated a spreadsheet on participant characteristics (e.g., sex, handedness) into the original data file, and uploaded individual trial data (see hyperlink, Pg. 8, Ln. 178).

Reviewer #2:

Reviewer’s comment/s: Although the authors added a brief section on spatial compatibility effects in the revised MS, they support their main interpretation using papers building on different paradigms with potentially different spatial relationships (e.g., participants using both hands). I am therefore unsure if these results can be generalized to the presented study. It has already been revealed that the stimulus hand used here elicits predominantly orthogonal compatibility effects and that purely imitative effects exist, but are rare, more modest in magnitude, and likely elicited by different stimulus. Other stimuli/contexts might not elicit significant compatibility effects at all. This is the reason why reporting strength of the effect (incom-com contrasts) in slow and fast bins is important (compatibility x time bin interaction). Were all incom-com differences in slow and fast bins significant? This could help the interpretation. For example, numerical difference in fast bin between incom and com trials for neutral condition seems very small (12ms) compared to the same difference in unsafe condition (49ms). Moreover, it would be easier on a reader to present these results in a graph rather than table.

Authors’ response: We have now statistically confirmed the difference in compatibility effects (compatible-incompatible difference) between slow and fast reaction time bins (Pg. 11, Ln. 238). While these differences appeared to vary between groups, this particular statistical effect (i.e., prime x compatibility x time bin) did not reach significance. Along these lines, we have now incorporated a figure (Fig 2; Pg. 11, Ln. 244-246) that more appropriately illustrates the compatibility effects including error bars that indicate the standard deviations, and how there was some degree of overlap between each of the groups in this regard. Please note, we have opted to also retain the table so the reader may observe the original mean values for compatible and incompatible stimulus-response conditions.

Reviewer #3:

In this manuscript the authors used an automatic imitation task preceded by a sentence comprehension priming task to test for the influence of COVID-related primes on imitation. An imitation effect was observed but no influence of the priming task.

I appreciate that this manuscript has already been reviewed and it’s galling to have to deal with a new set of comments after two rounds of review, so I leave it to the editor’s discretion to decide which of the following points are essential for the authors to address.

Introduction:

Reviewer’s comment/s: The focus on ‘advanced mammals’ in the first sentence seems unnecessary and inaccurate – a wide range of species show forms of imitation, including birds (Richards et al., 2009).

Authors’ response: We have adapted the opening statement to exclusively focus on imitation within humans, as well as the subsequent citations (Pg. 2, Ln. 15-16).

Reviewer’s comment/s: The introduction is missing discussion of studies which do not show an effect of social priming on imitation, including several recent and more highly powered studies (Genschow et al., 2022; see also the recent special issue on this topic by Cracco et al., 2022).

Authors’ response: We have incorporated the suggested citations including the new inclusion of Cracco et al. (2022) (Pg. 2, Ln. 37).

Reviewer’s comment/s: The rationale for the study doesn’t follow from the discussion of the BIS in the introduction. According to the introduction, the effects of the COVID-related primes should only be seen in participants with tendencies to implement avoidance behaviours (last paragraph of page 3) making it unclear why any effect of COVID-related primes on imitation should be found in a more general population sample?

Authors’ response: We have elaborated on our logic by reframing it within the context of situational primes (Pg. 3, Ln. 60-61). Indeed, it is relevant that the avoidance-type behaviours induced by potential infectious disease can also manifest following a mere situational prime as featured within the present study (Pg. 3, Ln. 50-54).

Moreover, aside from a theoretical perspective, it is worthwhile noting that the present study was motivated by a general interest in how the unprecedented scale of the current pandemic could influence human behaviour (Pg. 3, Ln. 41-46). In so doing, we could expand upon the catalogue of behaviours that have been known to be unexpectedly affected following the emergence of COVID-19.

Method:

Reviewer’s comment/s: Please provide a clearer statement of the statistical power the sample size afforded. The two studies cited here are a decade old and more recent studies have used far larger sample sizes (again see Cracco et al. special issue / editorial). Although you can’t estimate the likely size of the effect you are trying to detect, you can at least state what effect you were powered to detect (e.g. ‘a sample of 20 participants per group provides 80% power to detect an effect of at least d=0.9 at an alpha level of .05’).

Authors’ response: We have now incorporated a posteriori power analysis with a view to reporting the achieved statistical power (Pg. 10, Ln. 214-216). Please note, we have also retained the sensitivity analysis alluding to the minimum detectable effect size owing to a suggestion made by another reviewer.

Reviewer’s comment/s: Please provide a figure illustrating the stimuli/task.

Authors’ response: We have introduced a new figure that illustrates the transition across the protocol including the main stimulus-response task (Fig 1; Pg. 8, Ln. 166-169).

Reviewer’s comment/s: 20 trials per condition is a lot fewer than in most automatic imitation tasks. Please provide rationale in terms of the number of trials needed to obtain a robust estimate of response time in each condition, and also indicate the mean/SD number of trials per condition remaining once false starts, delayed responses, and error trials were removed; and discuss how this relates to the number of trials required for estimating RT.

Authors’ response: We concur with the argument of there being a low number of trials. However, it is relevant to consider the potential trade-off between ideally accumulating a large and more representative number of trials, and the pragmatics surrounding participant retention within a more remote participant-led data collection. We have attempted to clarify this point following the comments of a separate reviewer (Pg. 7-8, Ln. 160-162). At the same time, we do not know of any formal and statistically verified recommendation for the number of trials that are needed for this particular paradigm.

An initial coarse outlook of the data would seem to suggest that participants responded within a reasonable or typical range of reaction times (grand M = 548 ms), while we statistically replicated the classic automatic imitation effect (i.e., compatible < incompatible). However, in order to more precisely gauge whether the number of trials were adequately representative of the participants behaviour, we further inspected the reaction time data by calculating both the within-participant SD and median. Here, it is assumed that a more representative sample would produce a lower SD, while the mean and median reaction times would be somewhat similar to each other. Firstly, we found a mean within-participant SD = 120 ms (equating to a Coefficient of Variation (CoV) = 21%). Meanwhile, the below graphic indicates that the mean and median reaction times featured a highly similar trend. Thus, we infer that the low number of trials within the present study were still able to capture participants’ behaviour.

The fore mentioned outcomes were no doubt facilitated by the removal of trials that featured a false start, delay and/or error (Pg. 10, Ln. 216-217). It is important to clarify that this procedure led to the removal of very few trials as is indicated by the mean and SD of the number of trials (Pg. 10, Ln. 217-223; N.B., having stated the number of trials allocated to fast and slow time bins (see below response to comment), it is possible to aggregate them to infer the total number of trials removed).

Results:

Reviewer’s comment/s: Given the very low number of trials I don’t think the bin analysis is appropriate. At the very least the authors should report the mean/SD number of trials remaining within each bin.

Authors’ response: We have now reported on the mean and SD of the number of trials comprising the different time bins within each condition (Pg. 10, Ln. 217-223).

It would be helpful to add the magnitude of the compatibility effects to Table 1 (i.e. the mean/SE of the Incompatible-Compatible difference across participants), please.

Authors’ response: We have now incorporated a figure (Fig 2; Pg. 11, Ln. 243-246) that more appropriately reflects the compatibility effects in question.

Last paragraph of page 9 – this states that the dependent variable was the compatibility effect ‘Further inspection of the compatibility effects as a function of movement time bins’ but then compatibility is reported as a factor in the analysis – so is this actually analysing RT not compatibility effect? Please clarify.

Authors’ response: We have adapted the text to more explicitly communicate our meaning (Pg. 10, Ln. 230).

First lines of page 10 – the Bayes’ factor associated with this interaction is in favour of the null hypothesis, suggesting limited support for different magnitude compatibility effects across the slow and fast bins – please check and then reflect this in your discussion (if indeed, given my points about trial numbers above, you retain the bin analysis).

Authors’ response: We would like to thank the reviewer for this observation, and concur with their evaluation. As a result, we have more cautiously interpreted the reaction time bin analysis, and highlighted the failure of the Bayes Factor analysis to fully corroborate our suggestions (Pg. 15, Ln. 351). With this in mind, it is relevant to note that simultaneous frequentist and Bayesian statistical approaches do not always complement each other.

Discussion:

Reviewer’s comment/s: Page 14 – ‘the present compatibility effects unfolded within the slow rather than fast reaction time bins’ – this isn’t quite accurate, the effects were greater in the slow bins but they were still present in the fast bins? A follow-up analysis of the simple effects of compatibility at each time bin would help to address this point.

Authors’ response: We have amended the relevant text so that the reported compatibility effect was smaller, and not absent, within the fast time bins (Pg. 15, Ln. 349-351).

Reviewer’s comment/s: I think the conclusion should be qualified ‘did not modulate automatic imitation in this group of young UK adults’.

Authors’ response: We have qualified the concluding remarks as suggested (Pg. 16, Ln. 361-362).

---

## [Decision Letter · Decision Letter 3]

30 Mar 2023

PONE-D-22-16580R3Does the threat of COVID-19 modulate automatic imitation?PLOS ONE

Dear Dr. Roberts,

Thank you for submitting your manuscript to PLOS ONE. After careful consideration, we feel that it has merit but does not fully meet PLOS ONE’s publication criteria as it currently stands. Therefore, we invite you to submit a revised version of the manuscript that addresses the points raised during the review process.

ACADEMIC EDITOR: On the basis of the reviews, the manuscript will be accepted once the minor changes requested have been made.  Please ensure that your decision is justified on PLOS ONE’s publication criteria and not, for example, on novelty or perceived impact.

We look forward to receiving your revised manuscript.

Kind regards,

Merryn D. Constable

Academic Editor

PLOS ONE

Journal Requirements:

Reviewers' comments:

Reviewer's Responses to Questions

**Comments to the Author**

1. If the authors have adequately addressed your comments raised in a previous round of review and you feel that this manuscript is now acceptable for publication, you may indicate that here to bypass the “Comments to the Author” section, enter your conflict of interest statement in the “Confidential to Editor” section, and submit your "Accept" recommendation.

Reviewer #2: All comments have been addressed

Reviewer #3: (No Response)

2. Is the manuscript technically sound, and do the data support the conclusions?

Reviewer #2: Partly

Reviewer #3: Yes

3. Has the statistical analysis been performed appropriately and rigorously? 

Reviewer #2: Yes

Reviewer #3: Yes

4. Have the authors made all data underlying the findings in their manuscript fully available?

Reviewer #2: Yes

Reviewer #3: Yes

5. Is the manuscript presented in an intelligible fashion and written in standard English?

Reviewer #2: Yes

Reviewer #3: Yes

6. Review Comments to the Author

Reviewer #2: (No Response)

Reviewer #3: Please comment on the low number of trials per condition per time bin in the discussion.

Page 10 line 230: the dependent variable in this analysis is RT not incompatible-compatible difference, so this should say ‘further inspection of the response times…’.

Page 11, line 238: does this t test refer to the compatibility effect in the fast or in the slow time bins? Please provide the equivalent test for the other time bin.

7. PLOS authors have the option to publish the peer review history of their article (what does this mean?). If published, this will include your full peer review and any attached files.

Reviewer #2: No

Reviewer #3: No

---

## [Author Response · Author response to Decision Letter 3]

9 Apr 2023

*Please note, the references to page and line numbers are taken with respect to the manuscript version, ‘Revised Manuscript with Track Changes’.

Reviewer #2: No response

Reviewer #3:

Reviewer comment/s: Please comment on the low number of trials per condition per time bin in the discussion.

Authors’ response: We have now drawn attention to this pitfall including mention for future directions (Pg. 15, Ln. 349-353).

Reviewer comment/s: Page 10 line 230: the dependent variable in this analysis is RT not incompatible-compatible difference, so this should say ‘further inspection of response times…’.

Authors’ response: We thank the reviewer for this observation, and have subsequently altered the text (Pg. 10, Ln. 223).

Reviewer comment/s: Page 11, line 238: does this t test refer to the compatibility effect in the fast or in the slow time bins? Please provide the equivalent test for the other time bin.

Authors’ response: The t-test in question alludes to the compatible-incompatible reaction time difference for the slow compared to fast reaction time bins (Pg. 10, Ln. 229-231). Please note, while both bins indicated a compatibility effect (i.e., compatible < incompatible), the direct comparison between the slow and fast compatible-incompatible difference could more clearly highlight or decompose the purported Compatibility x Time Bin interaction (Pg. 10, Ln. 228-229).

---

## [Editor Report · Decision Letter 4]

12 Apr 2023

Does the threat of COVID-19 modulate automatic imitation?

PONE-D-22-16580R4

Dear Dr. Roberts,

We’re pleased to inform you that your manuscript has been judged scientifically suitable for publication and will be formally accepted for publication once it meets all outstanding technical requirements.

Kind regards,

Merryn D. Constable

Academic Editor

PLOS ONE
---

## [Editor Report · Acceptance letter]

17 Apr 2023

PONE-D-22-16580R4 

Does the threat of COVID-19 modulate automatic imitation? 

Dear Dr. Roberts:

I'm pleased to inform you that your manuscript has been deemed suitable for publication in PLOS ONE. Congratulations! Your manuscript is now with our production department. 

Kind regards, 

on behalf of

Dr. Merryn D. Constable 

Academic Editor

PLOS ONE